# Crude and adjusted comparisons of cesarean delivery rates using the Robson classification: A population-based cohort study in Canada and Sweden, 2004 to 2016

Giulia M. Muraca[1,2,3]*, K.S. Joseph[2,4], Neda Razaz[1], Linnea V. Ladfors[1], Sarka Lisonkova[2,4], Olof Stephansson[1,5]

1 Clinical Epidemiology Unit, Department of Medicine, Solna, Karolinska University Hospital, Karolinska Institutet, Eugeniahemmet, Stockholm, Sweden, 2 Department of Obstetrics and Gynaecology, University of British Columbia and the Children's and Women's Hospital and Health Centre of British Columbia, Vancouver, British Columbia, Canada, 3 Departments of Obstetrics and Gynecology and Health Research Methods, Evidence & Impact, Faculty of Health Sciences, McMaster University, Ontario, Canada, 4 School of Population and Public Health, University of British Columbia, Vancouver, British Columbia, Canada, 5 Division of Women's Health, Department of Obstetrics, Karolinska University Hospital, Stockholm, Sweden

* muracag@mcmaster.ca

**Data Availability Statement:** The Swedish Medical Birth Registry is a national dataset and considered to be public property. However, access to the data

## Abstract

### Background

The Robson classification has become a global standard for comparing and monitoring cesarean delivery (CD) rates across populations and over time; however, this classification does not account for differences in important maternal, fetal, and obstetric practice factors known to impact CD rates. The objectives of our study were to identify subgroups of women contributing to differences in the CD rate in Sweden and British Columbia (BC), Canada using the Robson classification and to estimate the contribution of maternal, fetal/infant, and obstetric practice factors to differences in CD rates between countries and over time.

### Methods and findings

We conducted a population-based cohort study of deliveries in Sweden (January 1, 2004 to December 31, 2016; n = 1,392,779) and BC (March 1, 2004 to April 31, 2017; n = 559,205). Deliveries were stratified into Robson categories and the CD rate, relative size of each group and its contribution to the overall CD rate were compared between the Swedish and the Canadian cohorts. Poisson and log-binomial regression were used to assess the contribution of maternal, fetal, and obstetric practice factors to spatiotemporal differences in Robson group-specific CD rates between Sweden and BC.

Nulliparous women comprised 44.8% of the study population, while women of advanced maternal age (≥35 years) and women with overweight/obesity (≥25 kg/m$^2$) constituted 23.5% and 32.4% of the study population, respectively. The CD rate in Sweden was stable at approximately 17.0% from 2004 to 2016 (p for trend = 0.10), while the CD rate increased in BC from 29.4% to 33.9% (p for trend < 0.001). Differences in CD rates between Sweden

is given only to researchers with permission from the Swedish Ethical Review Authority and after approval of the research plan by a Swedish National Board of Health and Welfare data manager. The data request may be sent to The Swedish National Board of Health and Welfare (https://www.socialstyrelsen.se/). The Canadian data were acquired from Perinatal Services British Columbia. Access to the Canadian data is available to researchers, health managers and decision-makers via a formal data request process administered by Perinatal Services British Columbia. More information on the data holdings and the request process is available at http://www.perinatalservicesbc.ca/our-services/data-surveillance/perinatal-data-registry/data-requests. All inferences, opinions, and conclusions drawn in this publication are those of the authors, and do not reflect the opinions or policies of Perinatal Services BC.

**Funding:** GMM is the recipient of a Canadian Institutes of Health Research (CIHR) Postdoctoral Fellowship (www.cihr-irsc.gc.ca/); NR, LVL, and OS are supported by the Swedish Research Council for Health, Working Life and Welfare (Forskningsrådet om Hälsa, Arbetsliv och Välfärd; www.forte.se) NR: 4-2702/2019; LVL: 2019-01275; OS: 2013-09298). NR is also supported by the CIHR. OS is also supported by the Strategic Research Program in Epidemiology at Karolinska Institutet (www.ki.se/en/research/sfoepi-grants). SL is supported by a Scholar Award from the Michael Smith Foundation for Health Research (www.msfhr.org); and KSJ is supported by an Investigator award from the BC Children's Hospital Research Institute (www.bcchr.ca). None of the authors have financial relationships with any organisations that might have an interest in the submitted work in the previous three years, nor do they have other relationships or activities that could appear to have influenced the submitted work. The funders had no role in study design, data collection and analysis, decision to publish, or preparation of the manuscript.

**Competing interests:** The authors have declared that no competing interests exist.

**Abbreviations:** ARR, adjusted rate ratio; BC, British Columbia; BCPDR, British Columbia Perinatal Database Registry; BMI, body mass index; CD, cesarean delivery; CI, confidence interval; MBR, Medical Birth Register; OR, odds ratio; RR, rate ratio; WHO, World Health Organization.

and BC varied by Robson group, for example, in Group 1 (nullipara with a term, single, cephalic fetus with spontaneous labor), the CD rate was 8.1% in Sweden and 20.4% in BC (rate ratio [RR] for BC versus Sweden = 2.52, 95% confidence interval [CI] 2.49 to 2.56, $p < 0.001$) and in Group 2 (nullipara, single, cephalic fetus, term gestation with induction of labor or prelabor CD), the rate of CD was 37.3% in Sweden and 45.9% in BC (RR = 1.23, 95% CI 1.22 to 1.25, $p < 0.001$). The effect of adjustment for maternal characteristics (e.g., age, body mass index), maternal comorbidity (e.g., preeclampsia), fetal characteristics (e.g., head position), and obstetric practice factors (e.g., epidural) ranged from no effect (e.g., among breech deliveries; Groups 6 and 7) to explaining up to 5.2% of the absolute difference in the CD rate (Group 2: adjusted CD rate in BC 40.7%, adjusted RR = 1.09, 95% CI 1.08 to 1.12, $p < 0.001$). Adjustment also explained a substantial fraction of the temporal change in CD rates among some Robson groups in BC. Limitations of the study include a lack of information on intrapartum details, such as labor duration as well as maternal and perinatal outcomes associated with the observed differences in CD rates.

## Conclusions

In this study, we found that several factors not included in the Robson classification explain a significant proportion of the spatiotemporal difference in CD rates in some Robson groups. These findings suggest that incorporating these factors into explanatory models using the Robson classification may be useful for ensuring that public health initiatives regarding CD rates are evidence informed.

## Author summary

### Why was this study done?

- The Robson classification system is a World Health Organization (WHO)-endorsed global standard for comparing and monitoring cesarean delivery (CD) rates.

- This classification does not include important maternal, fetal, and obstetric practice factors known to influence CD rates.

- The contribution of these characteristics to CD rate comparisons between and within populations has been identified as a key deficiency of this classification scheme and has not been comprehensively quantified.

### What did the researchers do and find?

- We conducted a population-based cohort study including 1,951,984 deliveries in Sweden and Canada between 2004 and 2016 to assess differences in CD rates among Robson groups, between countries and over time, with and without adjustment for maternal, fetal, and obstetric practice factors.

- The effect of adjustment between countries varied by Robson group from having no effect in some groups to explaining up to 61% of the variation in CDs in others.

- Adjustment for maternal, fetal, and obstetric practice factors explained a substantial fraction of the temporal change in CD rates among some Robson groups in Canada but had little impact on temporal changes in CD rates among Robson groups in Sweden.

### What do these findings mean?

- Public health initiatives based on Robson-classified CD rates may be misinformed without a comprehensive consideration of relevant maternal, obstetric practice, and fetal factors.

- Comprehensive and accurate perinatal data collection beyond the Robson criteria is necessary to ensure policies regarding CD rates are suitably evidence informed and prioritized.

- Future studies are warranted to evaluate the differences in CD rates in each Robson group in relation to maternal, fetal, and infant morbidity and mortality.

## Introduction

In 2015, the World Health Organization (WHO) endorsed the Robson classification system as a global standard for comparing and monitoring cesarean delivery (CD) rates across populations and over time [1]. As a result, the use of the Robson classification in evaluating CD trends has expanded over the last decade and become a common tool to inform obstetric practice worldwide [2–7]. This classification scheme stratifies deliveries into 10 mutually exclusive and all-inclusive categories based on 6 obstetric characteristics: parity (nulliparous versus parous), previous CD (yes/no), plurality (single versus multiple fetuses), fetal presentation (cephalic, breech, transverse/oblique), labor onset (spontaneous, induced, prelabor CD), and gestational age ($<37$ versus $\geq37$ weeks) [8,9].

The Robson classification system represents an elegant method that uses stratification (to control for confounding) in order to isolate the effect of specific obstetric practices on CD rates. Nevertheless, there are important determinants of spatial and temporal variations in CD rates that are not integrated into the Robson analysis scheme [10,11]. For example, maternal characteristics such as age [12] and body mass index (BMI) [13] are 2 important determinants of CD rates that are not addressed by the Robson strata. Similarly, obstetric practice factors, such as the clinical management of postterm delivery [14], as well as fetal factors, such as position of the fetal head at delivery [15] and fetal size [16], have also been associated with CD but are not considered in the Robson scheme. Inferences about spatiotemporal differences in CD rates may be biased if they are based on patterns within the Robson groups but without appropriate consideration of these extraneous factors that influence CD rates [5].

Statistical adjustment using regression techniques can address the contribution of extraneous determinants (such as maternal age and BMI) so that the true association between obstetric practice and CD rate can be quantified. The few studies that have applied regression to the assessment of CD rates using the Robson framework have been limited by single-center or cross-sectional study designs [17,18], while prior population-based or longitudinal studies lacked information on important risk factors for CD (e.g., labor onset, early-

pregnancy BMI, maternal comorbidity, fetal factors) [5–7,19–21]. As a result, the effect of maternal, fetal, and obstetric practice factors (not represented in the Robson classification) on international and temporal comparisons of CD rates using the Robson classification is unknown.

We carried out a study to identify subpopulations responsible for differences in CD rates between Sweden and British Columbia (BC), Canada using the Robson classification system. Further, we sought to quantify the contribution of maternal characteristics, obstetric practice factors, and fetal characteristics to variations in the CD rate (a) between countries; and (b) within each country over a 13-year period.

## Methods

This study is reported as per the Strengthening the Reporting of Observational Studies in Epidemiology (STROBE) guideline (S1 STROBE Checklist). The study methodology and analysis plan (S1 Text) were developed prior to data extraction and the actual analysis.

We conducted a population-based, retrospective, cohort study including all deliveries (live births and stillbirths) ≥22 completed weeks of gestation in Sweden and BC. Data for the Swedish cohort of deliveries between January 1, 2004 and December 31, 2016 were obtained from the Swedish Medical Birth Register (MBR), a validated, nationwide health register containing more than 98% of all births in Sweden [22]. Data for the Canadian cohort were obtained from the British Columbia Perinatal Database Registry (BCPDR) for the fiscal years 2004/2005 to 2016/2017 (hereafter referred to as years 2004 to 2016). The BCPDR contains information on approximately 99% of births in the province of BC [23]. Validation studies show that the BCPDR is an accurate and comprehensive source of perinatal information [24,25].

Both the MBR and the BCPDR contain detailed demographic and clinical information on all mothers and babies including information on diagnoses and procedures classified using standardized codes [22,23]. The study was restricted to the years 2004 to 2016 when all diagnostic codes conformed to the 10th revision of the International Statistical Classification of Diseases and Related Health Problems in Sweden (ICD-10-SE) [26] and Canada (ICD-10-CA) [27]. Similarly, procedure codes were consistently coded over the study period with the Nordic Classification of Care Measures [28] in Sweden and the Canadian Classification of Health Interventions in Canada [29].

### Analysis of CD rates using the Robson classification system

Data were grouped into Robson categories and modified from the 10-group system to a 12-group system, used commonly [9] to differentiate women in Groups 2 and 4 between those who had an intrapartum CD after induction (defined as 2a and 4a) and those who had a CD before labor onset (2b and 4b; Table 1). Women with twin (or higher order) pregnancies were only counted once; the mode of delivery of the second twin was used in the analysis. The overall CD rate, the relative size of each Robson group, and the absolute contribution of each group to the overall CD rate were compared between the Canadian and Swedish cohorts using the Standard Robson Classification Report Table. We also assessed the data quality, population distributions, and Robson group-specific CD rates against WHO-recommended benchmarks using standardized criteria [1]. Temporal trends in CD rates were described by Robson group and assessed using the Cochran–Armitage test. Temporal trends in the relative contribution of each Robson group to the overall CD rate were also described.

**Table 1.  Distribution of maternal, obstetric practice, and fetal/infant characteristics among deliveries in Sweden and BC, Canada, 2004–2016.**

| Maternal, obstetric practice, and fetal/infant characteristic | All deliveries (N = 1,951,984) | Sweden (n = 1,392,779) No. (%) | BC (n = 559,205) No. (%) | Standardized difference* |
|---|---|---|---|---|
| Maternal age (year) | | | | 0.17 |
| <20 | 29,166 (1.5) | 19,859 (1.4) | 9,307 (1.7) | |
| 20–24 | 234,269 (12.0) | 175,914 (12.6) | 58,355 (10.4) | |
| 25–29 | 548,949 (28.1) | 412,828 (29.6) | 136,121 (24.3) | |
| 30–34 | 676,342 (34.6) | 482,275 (34.6) | 194,067 (34.7) | |
| 35–39 | 373,699 (19.1) | 247,619 (17.8) | 126,080 (22.5) | |
| 40–44 | 84,413 (4.3) | 51,616 (3.7) | 32,797 (5.9) | |
| $\geq$45 | 5,146 (0.3) | 2,668 (0.2) | 2,478 (0.4) | |
| Maternal early-pregnancy BMI (kg/m$^2$) | | | | 0.63 |
| Underweight (<18.5) | 50,221 (2.6) | 28,394 (2.0) | 21,827 (3.9) | |
| Normal weight (18.5–24.9) | 1,000,635 (51.3) | 759,464 (54.5) | 241,171 (43.1) | |
| Overweight (25.0–29.9) | 419,302 (21.5) | 333,738 (24.0) | 85,564 (15.3) | |
| Obese class I (30.0–34.9) | 144,819 (7.4) | 112,532 (8.1) | 32,287 (5.8) | |
| Obese class II (35.0–39.9) | 48,684 (2.5) | 35,785 (2.6) | 12,899 (2.3) | |
| Obese class III ($\geq$40.0) | 19,946 (1.0) | 12,911 (0.9) | 7,035 (1.3) | |
| Missing | 268,377 (13.7) | 109,955 (7.9) | 158,422 (28.3) | |
| Parity | | | | 0.04 |
| 0 | 875,011 (44.8) | 615,451 (44.2) | 259,560 (46.4) | |
| 1 | 713,765 (36.6) | 512,130 (36.8) | 201,635 (36.1) | |
| 2 | 252,053 (12.9) | 185,055 (13.3) | 66,998 (12.0) | |
| 3–4 | 93,654 (4.8) | 67,496 (4.8) | 26,158 (4.7) | |
| $\geq$5 | 17,319 (0.9) | 12,645 (0.9) | 4,674 (0.8) | |
| Missing | <185 (0.0) | <5 (0.0) | 180 (0.0) | |
| Smoking during pregnancy | 143,148 (7.3) | 95,314 (6.8) | 47,834 (8.6) | 0.06 |
| Preexisting diabetes | 9,660 (0.5) | 6,715 (0.5) | 2,945 (0.5) | 0.01 |
| Preeclampsia/eclampsia | 47,990 (2.5) | 39,509 (2.8) | 8,481 (1.5) | −0.09 |
| Chronic hypertension | 13,754 (0.7) | 10,000 (0.7) | 3,754 (0.7) | −0.01 |
| In vitro fertilization | 50,015 (2.6) | 39,590 (2.8) | 10,425 (1.9) | −0.07 |
| Onset of labor | | | | 0.28 |
| Spontaneous | 1,410,549 (72.3) | 1,049,748 (75.4) | 360,801 (64.5) | |
| Induced | 331,153 (17.0) | 213,499 (15.3) | 117,654 (21.0) | |
| CD before labor | 202,714 (10.4) | 121,978 (8.8) | 80,736 (14.4) | |
| Unknown | 7,568 (0.4) | 7,554 (0.5) | 14 (0.0) | |
| Gestational age (completed weeks) | | | | 0.29 |
| Very early preterm (22–27) | 7,062 (0.4) | 4,663 (0.3) | 2,399 (0.4) | |
| Early preterm (28–31) | 11,677 (0.6) | 7,695 (0.6) | 3,982 (0.7) | |
| Late preterm (32–36) | 107,720 (5.5) | 63,490 (4.6) | 44,230 (7.9) | |
| Term (37–41) | 1,723,126 (88.3) | 1,221,617 (87.7) | 501,509 (89.7) | |
| Postterm ($\geq$42) | 101,377 (5.2) | 94,880 (6.8) | 6,497 (1.2) | |
| Missing | 1,022 (0.1) | 434 (0.0) | 588 (0.1) | |
| Epidural anesthesia | 601,490 (30.8) | 426,192 (30.6) | 175,298 (31.3) | 0.02 |
| Vacuum | 137,204 (7.0) | 98,043 (7.0) | 39,161 (7.0) | 0.00 |
| Forceps | 19,547 (1.0) | 2,444 (0.2) | 17,103 (3.1) | 0.23 |
| Plurality | | | | 0.08 |
| Singleton | 1,923,311 (98.5) | 1,373,078 (98.6) | 550,233 (98.4) | |
| Multiple | 28,673 (1.5) | 19,701 (1.4) | 8,972 (1.6) | |

(*Continued*)

**Table 1.** (Continued)

| Maternal, obstetric practice, and fetal/infant characteristic | All deliveries (N = 1,951,984) | Sweden (n = 1,392,779) No. (%) | BC (n = 559,205) No. (%) | Standardized difference* |
|---|---|---|---|---|
| Infant birth weight (g) | | | | 0.20 |
| <2,500 | 80,830 (4.1) | 52,562 (3.8) | 28,268 (5.1) | |
| 2,500–2,999 | 231,533 (11.9) | 148,620 (10.7) | 82,913 (14.8) | |
| 3,000–3,499 | 651,997 (33.4) | 445,767 (32.0) | 206,230 (36.9) | |
| 3,500–3,999 | 657,834 (33.7) | 484,302 (34.8) | 173,532 (31.0) | |
| 4,000–4,499 | 266,752 (13.7) | 209,206 (15.0) | 57,546 (10.3) | |
| ≥4,500 | 60,728 (3.1) | 50,385 (3.6) | 10,343 (1.8) | |
| Missing | 2,310 (0.1) | 1,937 (0.1) | 373 (0.1) | |
| Infant head circumference at birth (cm) | | | | 0.16 |
| <33 | 120,738 (6.2) | 77,129 (5.5) | 43,609 (7.8) | |
| 33–34 | 610,852 (31.3) | 418,513 (30.0) | 192,339 (34.4) | |
| 35–36 | 913,013 (46.8) | 661,093 (47.5) | 251,920 (45.0) | |
| ≥37 | 270,225 (13.8) | 204,664 (14.7) | 65,561 (11.7) | |
| Missing | 37,156 (1.9) | 31,380 (2.3) | 5,776 (1.0) | |
| Fetal head in occiput posterior position at delivery | 92,042 (4.7) | 59,495 (4.3) | 32,547 (5.8) | 0.07 |
| Congenital anomaly | 77,042 (3.9) | 48,686 (3.5) | 28,356 (5.1) | 0.08 |

*Standardized difference values >0.1 are considered indicative of a significant difference between groups.

Small numbers (<5) were suppressed in order to prevent potential identification and breach of confidentiality (cells were sometimes suppressed to prevent back calculation).

BC, British Columbia; BMI, body mass index; CD, cesarean delivery; No., number.

## Contribution of maternal, fetal, and obstetric practice factors to spatial differences in CD

We categorized potential determinants of CD into 4 groups: maternal characteristics, maternal comorbidity, factors related to obstetric practice, and fetal factors. The maternal characteristics included age (<20, 20 to 24, 25 to 29, 30 to 34, 35 to 39, 40 to 44, ≥45 years), parity (in parous groups; 1, 2, 3 to 4, ≥5), early-pregnancy BMI classified in kg/m$^2$ as normal (18.5 to 24.9), underweight (<18.5), overweight (25.0 to 29.9), obesity class 1 (30.0 to 34.9), class 2 (35.0 to 39.9), and class 3 (≥40), and maternal smoking during pregnancy. In Sweden and BC, maternal BMI in early pregnancy was calculated from self-reported height and weight or provider assessment at the first antenatal visit, which typically occurs within the first 14 weeks of gestation [22,23]. We also adjusted for maternal comorbidity, which included chronic hypertension, early-pregnancy diabetes, in vitro fertilization, and preeclampsia/eclampsia. Epidural anesthesia and postterm delivery (≥42 completed weeks gestation) were the obstetric practice factors identified as potential determinants of CD. We chose these 2 obstetric practice factors because (a) we hypothesized they would have substantial variability between the 2 countries and over time; and (b) this information was reliably captured in both the MBR and the BCPDR. Finally, we also adjusted for fetal/infant characteristics: position of the fetal head at delivery (in groups with a cephalic-presenting fetus; occiput posterior versus occiput anterior), birth weight, head circumference, and congenital anomaly.

The distribution of these risk factors for CD were compared between Sweden and BC using frequencies and standardized differences [30]. Modified Poisson regression with robust standard errors was used to estimate the effect of maternal characteristics, comorbidity, fetal characteristics, and obstetric practice factors on the differences between CD rates in each Robson

group in BC versus Sweden. These analyses were performed on individual pregnancies, which made it possible for a woman to contribute more than 1 pregnancy in the Robson groups that included multiparous women. Therefore, we used generalized estimating equations (with the mother's identification as a cluster and assuming an exchangeable correlation structure), with adjustment for the possible correlation in outcome that could be introduced by subsequent births by the same mother. The difference between CD rates in the 2 countries that was explained by differences in risk factors was assessed by comparing the crude rate ratio (RR) and adjusted RR (ARR) in each Robson group and on the absolute scale, by comparing the crude and adjusted CD rate in BC versus Sweden. Missing BMI values were modeled using the missing-indicator method. Records with missing values for other variables were excluded as they did not exceed 3% of the total study population; thus, we conducted a complete-case analysis with respect to these variables (Fig 1).

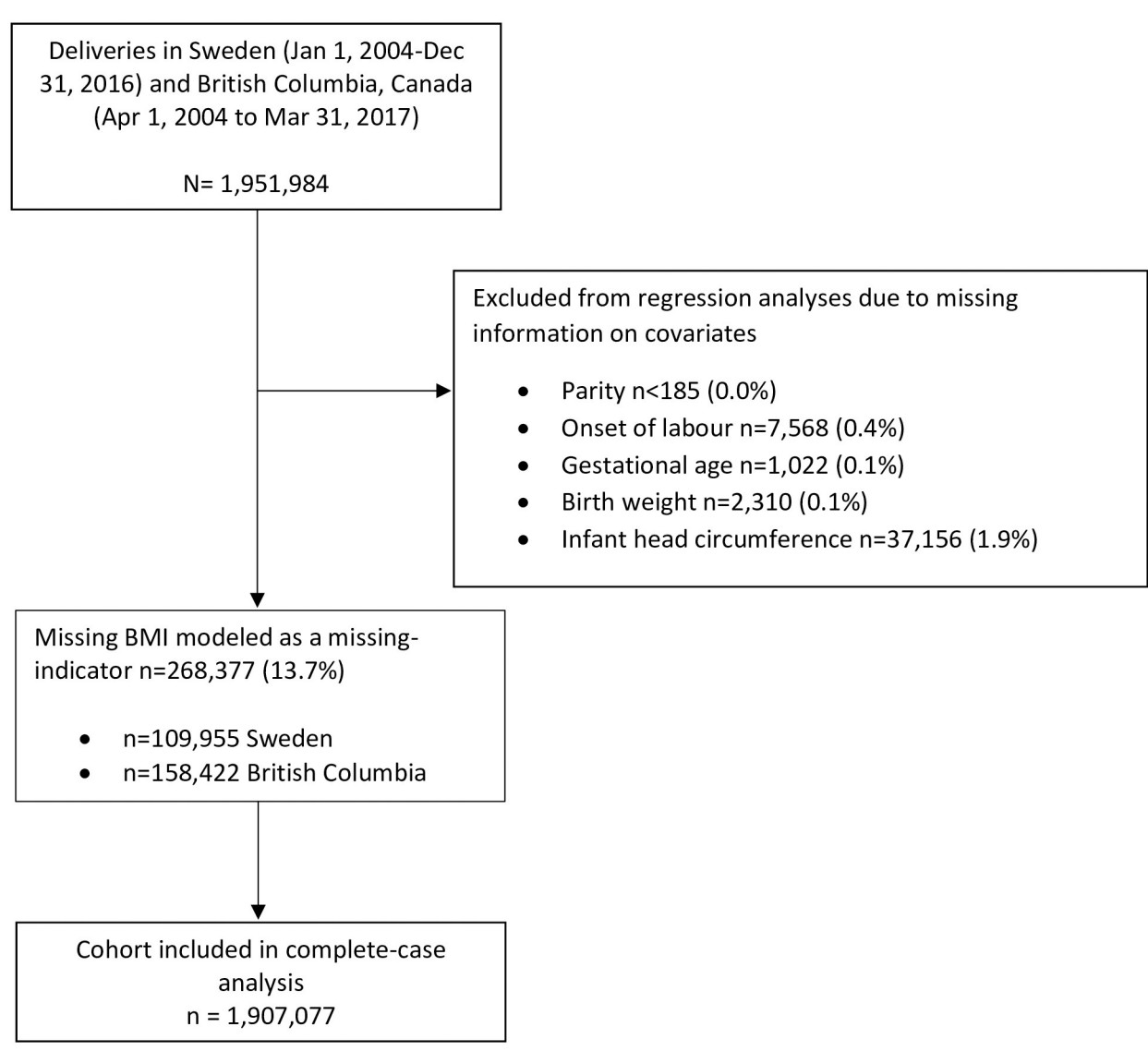

**Fig 1. Flow diagram showing derivation of cohort.**

We calculated RRs instead of odds ratios (ORs) in our spatial analyses of Robson group-specific CD rates, as the RR is more readily interpretable and the OR does not correspond to the RR when the event rate is relatively high. However, the RR is artefactually constrained when the baseline CD rate is high (e.g., cannot exceed 2 if the baseline CD rate is 55%). The OR is not susceptible to this constraint, and therefore, we also calculated ORs and 95% confidence intervals (CIs) for Robson group-specific CD rates in BC versus Sweden.

## Contribution of maternal, fetal, and obstetric practice factors on temporal changes in CD

We quantified the temporal trend in CD rates in Sweden and BC by year and also by period (2004 to 2007, 2008 to 2010, 2011 to 2013, and 2014 to 2016) in the 3 Robson groups with the largest contributions to the overall CD rate. The magnitude of the change in CD rates over time was estimated using RRs and 95% CIs.

To evaluate the contribution of changes in maternal, fetal, and obstetric practice factors to temporal changes in CD rates in these Robson groups, we first compared the temporal trends in the risk factors listed above by period (2014 to 2016 versus 2004 to 2007) using RRs and 95% CIs (the linear trend by year was assessed using the Cochran–Armitage test). Secondly, since changes in maternal characteristics (e.g., increases in advanced maternal age) and maternal comorbidity (e.g., diabetes) can lead to changes in obstetric practice factors (e.g., increases in epidural use), we used sequential log-binomial models to identify the effect of each group of factors on the relationship between year or period and CD. The sequential approach was carried out by fitting a series of models with additional groups of factors added to each model in the sequence outlined above to quantify the contribution of each group of factors to CD trends over time. Temporal trends were estimated both for individual year and period through separate regression models.

## Sensitivity analyses

Antenatal and maternity care in Sweden and Canada is universal and free; however, the model of maternity care differs substantially between countries. In Sweden, midwives are the primary care providers for women with low-risk pregnancies [31], whereas in BC, midwives are involved in maternity care for about 25.3% of deliveries [32]. To address differences in maternity care models in each country, our first sensitivity analysis compared deliveries in Sweden with those in BC restricted to women with midwifery care.

We performed 2 additional sensitivity analyses to address missing information about early-pregnancy BMI (13.8% of the study population; 7.9% and 28.3% among deliveries in Sweden and BC, respectively). First, we conducted complete-case regression analysis by excluding deliveries without data on BMI from the regression models. Second, missing values on BMI were estimated using a Markov Chain Monte Carlo multiple imputation method. Results from 10 multiple imputation cycles were combined with the use of PROC MIANALYZE in SAS.

The a priori level of statistical significance was set at a 2-sided $p$-value $< 0.05$ for all analyses. The Cochran–Armitage test was used to test for significance of linear trend, and the Wald chi-square test was used to test significance of differences in CD rates in the crude and adjusted regression models. All analyses were conducted using SAS version 9.4 (SAS Institute, Cary, North Carolina, United States of America). The study was approved by the Clinical Research Ethics Board at the University of British Columbia (H14-00674) and the Research Ethics Committee at Karolinska Institutet, Stockholm, Sweden (No. 2008/1182-31/4).

## Results

The study population included 1,392,779 deliveries in Sweden and 559,205 deliveries in BC from 2004 to 2016. Nulliparous women comprised 44.8% of the study population, while women of advanced maternal age ($\geq$35 years) and women with overweight or obesity ($\geq$25 kg/m$^2$) constituted 23.5% and 32.4% of the study population, respectively. Inductions occurred in 17.0% of deliveries, while 10.4% were CD before labor. Approximately 6.5% of deliveries occurred at preterm gestation (<37 weeks) and the proportion of deliveries that followed a multifetal pregnancy (twins or higher order) was 1.5% (Table 1). The overall rate of CD was 17.3% in Sweden and 31.2% in BC (Table 2). The rate of CD was higher in BC compared with Sweden in all Robson groups except in Group 4 (parous women with a single, cephalic fetus at term gestation with an induction or CD prior to labor) and among deliveries with a transverse/oblique-presenting fetus (Group 9). The standard assessment of the data quality, obstetric population distribution, and Robson group-specific CD rates is included in S1 Table. The largest differences in the CD rate between BC and Sweden were among women with at least 1 previous CD and a single, cephalic fetus at term gestation (Robson Group 5, OR in BC versus Sweden 4.09, 95% CI 4.00 to 4.18, $p$ < 0.001) and among nulliparous women with a single, cephalic fetus at term gestation and spontaneous onset of labor (Group 1, OR 2.91, 95% CI 2.87 to 2.96, $p$ < 0.001; S2 Table).

**Table 2. The Robson Classification Report Table [9], Sweden and BC, Canada, 2004–2016.**

| Robson group | CDs No. | | All deliveries No. | | Relative size (%) | | CD rate (%) | | Absolute contribution to overall CD rate (%) | | Relative contribution to overall CD rate (%) | |
|---|---|---|---|---|---|---|---|---|---|---|---|---|
| | Sweden | BC | Sweden | BC | Sweden | BC | Sweden | BC | Sweden | BC | Sweden | BC |
| 1. Nulliparous, singleton, cephalic, $\geq$37 weeks, spontaneous labor | 34,884 | 30,848 | 431,199 | 151,106 | 31.0 | 27.0 | 8.1 | 20.4 | 2.5 | 5.5 | 14.5 | 17.7 |
| 2. Nulliparous, singleton, cephalic, $\geq$37 weeks, induced or CD before labor | 42,474 | 31,255 | 113,927 | 68,084 | 8.2 | 12.2 | 37.3 | 45.9 | 3.0 | 5.6 | 17.6 | 17.9 |
| 2a. Induced | 26,623 | 24,491 | 98,076 | 61,320 | 7.0 | 11.0 | 27.2 | 39.9 | 1.9 | 4.4 | 11.0 | 14.0 |
| 2b. CD before labor | 15,851 | 6,764 | 15,851 | 6,764 | 1.1 | 1.2 | 100.0 | 100.0 | 1.1 | 1.2 | 6.6 | 3.9 |
| 3. Parous, singleton, cephalic, $\geq$37 weeks, no previous CD, spontaneous labor | 8,114 | 3,910 | 500,236 | 151,071 | 35.9 | 27.0 | 1.6 | 2.6 | 0.6 | 0.7 | 3.4 | 2.2 |
| 4. Parous, singleton, cephalic, $\geq$37 weeks, no previous CD, induced or CD before labor | 21,296 | 5,328 | 98,901 | 40,740 | 7.1 | 7.3 | 21.5 | 13.1 | 1.5 | 1.0 | 8.8 | 3.1 |
| 4a. Induced | 4,461 | 2,656 | 82,066 | 38,068 | 5.9 | 6.8 | 5.4 | 7.0 | 0.3 | 0.5 | 1.8 | 1.5 |
| 4b. CD before labor | 16,835 | 2,672 | 16,835 | 2,672 | 1.2 | 0.5 | 100.0 | 100.0 | 1.2 | 0.5 | 7.0 | 1.5 |
| 5. Parous, singleton, cephalic, $\geq$37 weeks, with a previous CD | 61,923 | 58,277 | 120,104 | 71,665 | 8.6 | 12.8 | 51.6 | 81.3 | 4.5 | 10.4 | 25.7 | 33.4 |
| 6. Nulliparous, singleton, breech | 25,804 | 12,351 | 27,503 | 12,932 | 2.0 | 2.3 | 93.8 | 95.5 | 1.9 | 2.2 | 10.7 | 7.1 |
| 7. Parous, singleton, breech | 14,661 | 7,908 | 16,572 | 8,771 | 1.2 | 1.6 | 88.5 | 90.2 | 1.1 | 1.4 | 6.1 | 4.5 |
| 8. Multiple pregnancy (twins or higher-order multiples) | 10,780 | 6,244 | 19,701 | 8,972 | 1.4 | 1.6 | 54.7 | 69.6 | 0.8 | 1.1 | 4.5 | 3.6 |
| 9. Singleton, transverse or oblique lie | 1,892 | 1,731 | 1,905 | 1,828 | 0.1 | 0.3 | 99.3 | 94.7 | 0.1 | 0.3 | 0.8 | 1.0 |
| 10. Singleton, cephalic, <37 weeks | 17,178 | 11,699 | 58,500 | 37,968 | 4.2 | 6.8 | 29.4 | 30.8 | 1.2 | 2.1 | 7.1 | 6.7 |
| Unknown* | 2,130 | 4,904 | 4,231 | 6,068 | 0.3 | 1.0 | 50.3 | 80.8 | 0.2 | 0.9 | 0.9 | 2.8 |
| Total obstetric population | 241,136 | 174,455 | 1,392,779 | 559,205 | 100.0 | 100.0 | 17.3 | 31.2 | 17.3 | 31.2 | 100 | 100 |

*All remaining records that could not be classified due to missing information on 1 or more of the following variables: fetal presentation, parity, gestational age, type of labor, or previous CD.

BC, British Columbia; CD, cesarean delivery; No., number.

## Temporal trends in CD rates by Robson group

In Sweden, the overall rate of CD remained stable from 16.8% in 2004 to 17.6% in 2016 (P for trend = 0.1; S3 Table). Notable changes by Robson group included a decline in CDs among nulliparous women with a term, singleton fetus in cephalic presentation with induced labor (Group 2a; 28.6% to 24.3%), and an increase in CD from 49.4% to 52.8% among women with a previous CD (Group 5). In contrast, CD rates increased substantially during this period in BC. Overall, the rate of CD increased from 29.4% to 34.0% (P for trend < 0.0001; S4 Table). The CD rate in BC increased from 37.5% to 45.4% in Robson Group 2a and from 26.7% to 37.6% in women who delivered preterm (Group 10). Temporal trends in the rate of CD, the relative size, and the relative contribution to the overall CD rate among select Robson groups of interest are compared in Fig 2 in Sweden and BC.

## Contribution of each Robson group to the overall rate of CD

In both Swedish and BC cohorts, the same 3 Robson groups showed the largest contribution to the overall rate of CD (Table 2). The largest contributing group was Robson Group 5, women with at least 1 previous CD and a term, singleton, cephalic-presenting fetus. Although this group accounted for only 8.6% and 12.8% of the total obstetric population in Sweden and BC, respectively, the high rate of CD in these women (51.6% in Sweden; 81.3% in BC) made this group responsible for approximately 1 in 4 CDs in Sweden and 1 in 3 CDs in BC. Robson Group 1 (nulliparous women with a single, cephalic-presenting fetus at term gestation, and spontaneous labor) made the second largest absolute contribution to the overall rate of CD (2.5% in Sweden and 5.5% in BC). Group 2a (nulliparous women with a single, cephalic-presenting fetus at term gestation, and induced labor) made the third largest absolute contribution to the overall CD rate (1.9% in Sweden and 4.4% in BC). CDs in the abovementioned Robson groups (5, 1, and 2a) were responsible for 82.1% (11.5% of 14.0%) of the excess CDs in BC compared with Sweden (Fig 3).

## Contribution of maternal, fetal, and obstetric practice factors to spatial differences in CD rates

The distribution of risk factors for CD in Sweden and BC by Robson group are included in S5–S16 Tables. In general, deliveries to women with advanced maternal age, labor induction, and CD before labor were more common in BC, while spontaneous onset of labor and macrosomic infants were more common in Sweden (Table 1). The rate of preeclampsia was higher in Sweden compared with BC in all Robson groups (S1 Fig), and the proportion of women delivering at 42 weeks' gestation or beyond was higher among Swedish women, particularly in Group 2a (31.4% in Sweden versus 3.0% in BC). However, most women who delivered at post-term gestation in Sweden delivered by 42 + 2 (50%) and 42 + 3 (75%) weeks' gestation.

After adjustment for maternal, fetal, and obstetric practice factors not represented in the Robson scheme, the rate of CD in BC compared with Sweden was significantly attenuated in Groups 1, 2, 4, 5, and 8 and significantly increased in Group 10 (Table 3). Adjustment explained 14% of the increase in CD in BC versus Sweden in Robson Group 2 (RR = 1.23, 95% CI 1.22 to 1.25, $p < 0.001$; ARR = 1.09, 95% CI 1.08 to 1.12, $p < 0.001$) and 10.0% of the increase in CD in BC versus Sweden in Robson Group 5 (RR = 1.58, 95% CI 1.57 to 1.59, $p < 0.001$; ARR = 1.48, 95% CI 1.47 to 1.50, $p < 0.001$), which corresponded to a 5% absolute reduction in the adjusted CD rate in BC in both groups. Notably, the difference in the rate of CD between BC and Sweden widened by 30.8% after adjustment in Group 4, the only group in which the crude rate of CD was substantially lower in BC compared with Sweden (Table 3).

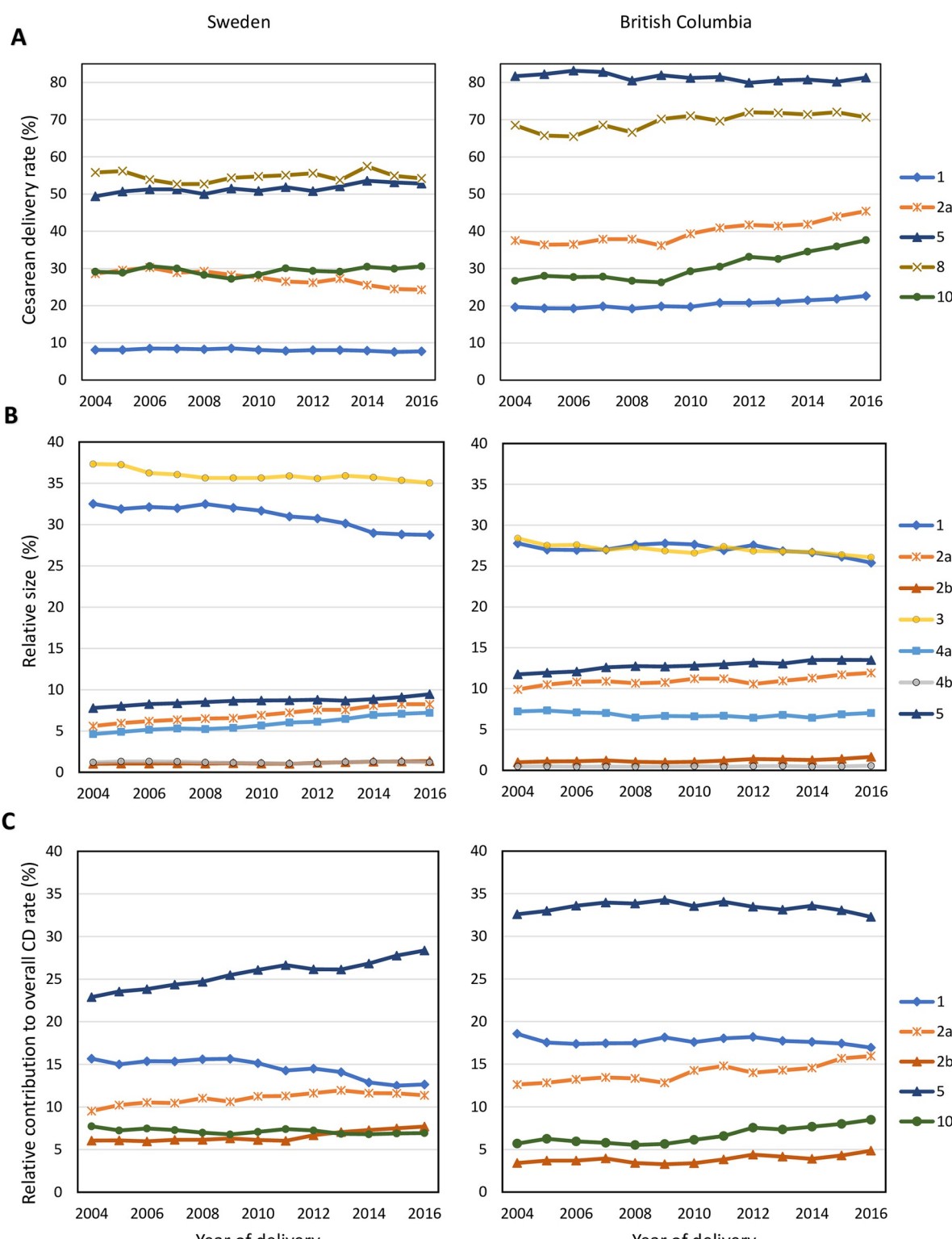

**Fig 2. Rate of CD (panel A), relative size (panel B), and contribution to the overall CD rate (panel C) among selected Robson groups, Sweden, and BC, Canada, 2004–2016.** BC, British Columbia; CD, cesarean delivery.

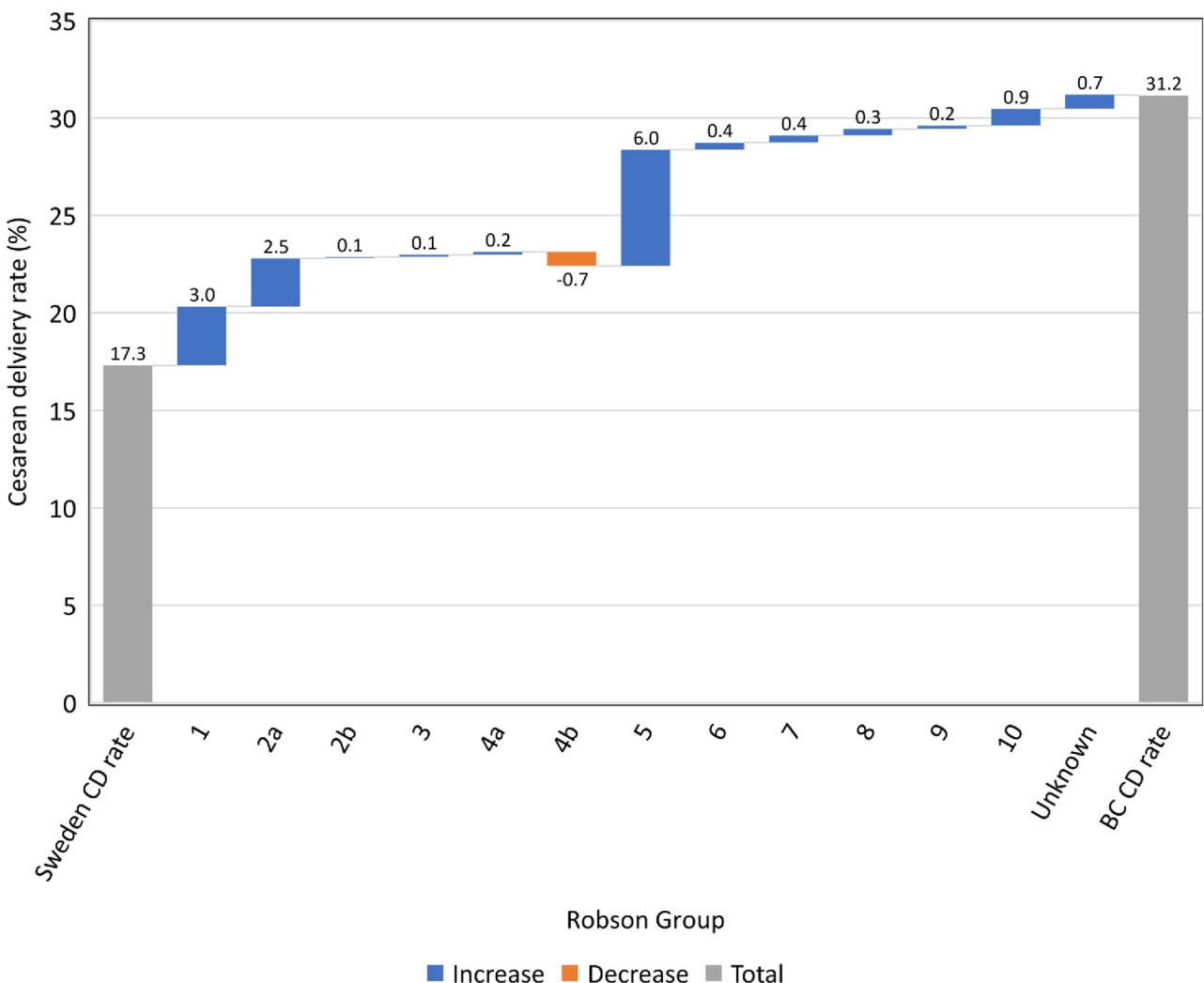

**Fig 3. Cumulative difference in rate of CD (per 100 deliveries) in BC, Canada vs. Sweden, by Robson group, 2004–2016.** BC, British Columbia; CD, cesarean delivery.

### Contribution of maternal, fetal, and obstetric practice factors to temporal changes in CD rates

Changes in maternal, fetal, and obstetric practice factors over the study period are presented for Robson Groups 1, 2a, and 5, as these groups contributed most to the overall CD rates in BC and Sweden (S17–S19 Tables and S2–S4 Figs). Early-pregnancy BMI increased and smoking rates declined across all groups of women in both countries. Advanced maternal age (35 + years) increased, particularly in BC (e.g., from 19.4% in 2004 to 25.8% in 2016 in Robson Group 2a; S18 Table), while the rate of preeclampsia in Sweden decreased significantly in all 3 groups over the study period.

The crude CD rate among women in Robson Group 1 declined in Sweden from 8.1% in 2004 to 7.7% in 2016 (RR 2014 to 2016 versus 2004 to 2007 = 0.93, 95% CI 0.91 to 0.96, $p < 0.001$), and this rate was further attenuated after adjustment for maternal, infant, and obstetric practice factors (Table 4). In BC, the crude rate of CD in Robson Group 1 increased

**Table 3. Crude and adjusted rates and rate ratios for CD in BC vs. Sweden by Robson group, 2004–2016.**

| Robson group | Crude CD rate (%) | | CD BC vs. Sweden | | | | Adjusted CD rate in BC (%)[‡] |
|---|---|---|---|---|---|---|---|
| | Sweden | BC | RR (95% CI) | *P*-value[*] | ARR[†] (95% CI) | *P*-value[*] | |
| 1 | 8.1 | 20.4 | 2.52 (2.49–2.56) | <0.001 | 2.32 (2.29–2.36)[a,b] | <0.001 | 18.8 |
| 2 | 37.3 | 45.9 | 1.23 (1.22–1.25) | <0.001 | 1.09 (1.08–1.12)[b] | <0.001 | 40.7 |
| 3 | 1.6 | 2.6 | 1.60 (1.54–1.66) | <0.001 | 1.54 (1.48–1.60)[a] | <0.001 | 2.5 |
| 4 | 21.5 | 13.1 | 0.61 (0.59–0.62) | <0.001 | 0.49 (0.48–0.51) | <0.001 | 10.5 |
| 5 | 51.6 | 81.3 | 1.58 (1.57–1.59) | <0.001 | 1.48 (1.47–1.50) | <0.001 | 76.4 |
| 6 | 93.8 | 95.5 | 1.02 (1.01–1.02) | <0.001 | 1.02 (1.01–1.02)[b,c] | 0.02 | 95.5 |
| 7 | 88.5 | 90.2 | 1.02 (1.01–1.03) | <0.001 | 1.02 (1.01–1.03)[c] | 0.03 | 90.2 |
| 8 | 54.7 | 69.6 | 1.27 (1.25–1.30) | <0.001 | 1.19 (1.17–1.22)[c] | <0.001 | 65.1 |
| 9 | 99.3 | 94.7 | 0.95 (0.94–0.96) | <0.001 | 0.95 (0.94–0.97)[c] | 0.03 | 94.7 |
| 10 | 29.4 | 30.8 | 1.04 (1.03–1.07) | <0.001 | 1.11 (1.09–1.14)[d] | <0.001 | 32.6 |
| All groups | 17.3 | 31.2 | 1.80 (1.79–1.81) | <0.001 | | | |

[*]*P*-values represent significance of Wald chi-square test; the a priori level of statistical significance was set at a 2-sided *p*-value < 0.05.

[†]Adjusted models included maternal age, parity, early-pregnancy BMI, smoking during pregnancy, chronic hypertension, preexisting diabetes, in vitro fertilization, preeclampsia/eclampsia, postterm delivery, position of the fetal head at delivery, infant birth weight, infant head circumference, and congenital anomaly.

[‡]Adjusted CD rate in BC = crude CD rate in Sweden*ARR.

[a]Adjusted model also included epidural anesthesia.

[b]Adjusted model excluded parity due to group restriction to nulliparas.

[c]Adjusted model excluded position of the fetal head at delivery due to group restriction to non-cephalic fetal presentation or multiple gestation.

[d]Adjusted model excluded postterm delivery due to group restriction to preterm deliveries.

ARR, adjusted rate ratio; BC, British Columbia; BMI, body mass index; CD, cesarean delivery; CI, confidence interval; RR, rate ratio.

from 19.7% in 2004 to 22.6% in 2016 (RR 2014 to 2016 versus 2004 to 2007 = 1.12, 95% CI 1.09 to 1.16, *p* < 0.001). The observed increase in CD was entirely explained by adjustment for changes in maternal characteristics and obstetric practice factors (ARR 2014 to 2016 versus 2004 to 2007 = 1.01, 95% CI 0.99 to 1.04, *p* = 0.80; Table 4 and Fig 4).

Among women in Robson Group 2a, the crude rate of CD decreased from 28.6% in 2004 to 24.3% in 2016 in Sweden and increased from 37.5% to 45.4% in BC over the same period. Adjustment did not affect the trend in the rate of CD in this group in Sweden (Fig 4). In BC, adjustment for changes in maternal characteristics (mainly age) explained 7% of the 18% relative increase in CD over this period (Table 4). Nevertheless, the upward trend in CD by period remained significant after sequential adjustment for all determinants (ARR 2014 to 2016 versus 2004 to 2007 = 1.09, 95% CI 1.07 to 1.13, *p* < 0.001; Table 4).

In Sweden, the crude rate of CD in Robson Group 5 increased from 49.4% in 2004 to 52.8% in 2016 (RR 2014 to 2016 versus 2004 to 2007 = 1.05, 95% CI 1.03 to 1.06, *p* < 0.001) and the increase was unaffected by adjustment (Table 4 and Fig 4). The effect of adjustment for each group of risk factors by period in Groups 1, 2a, and 5 is tabulated for both cohorts in S20 Table.

## Sensitivity analyses

After restricting deliveries in BC to those with midwifery care, the overall rate of CD in BC was 19.9%. The main differences in the rates of CD in this restricted group compared with all deliveries in BC was a lower rate of CD in women with a previous CD (Group 5), women with multiple pregnancies (Group 8), and those who delivered preterm (Group 10; S21 Table). Among these women as well, adjustment for maternal, fetal, and obstetric practice factors

**Table 4. Unadjusted and adjusted rate ratios for CD in 2014–2016 vs. 2004–2007 among women in Robson Groups 1, 2a, and 5 after sequential adjustment\* for maternal characteristics, obstetric practice factors, and fetal/infant characteristics, Sweden and BC, Canada.**

| Determinants adjusted for | Robson Group 1 | | | | Robson Group 2a | | | | Robson Group 5 | | | |
|---|---|---|---|---|---|---|---|---|---|---|---|---|
| | Sweden | | BC | | Sweden | | BC | | Sweden | | BC | |
| | RR (95% CI) | P-value[†] | RR (95% CI) | P-value[†] | RR (95% CI) | P-value[†] | RR (95% CI) | P-value[†] | RR (95% CI) | P-value[†] | RR (95% CI) | P-value[†] |
| Unadjusted | 0.93 (0.91–0.96) | <0.001 | 1.12 (1.09–1.16) | <0.001 | 0.84 (0.82–0.87) | <0.001 | 1.18 (1.15–1.21) | <0.001 | 1.05 (1.03–1.06) | <0.001 | 0.98 (0.97–0.99) | 0.01 |
| Adjusted for maternal characteristics[a] | 0.92 (0.90–0.95) | <0.001 | 1.06 (1.03–1.09) | <0.001 | 0.84 (0.82–0.87) | <0.001 | 1.11 (1.08–1.14) | <0.001 | 1.05 (1.03–1.06) | <0.001 | 0.98 (0.97–0.99) | 0.03 |
| Also adjusted for maternal conditions[b] | 0.93 (0.90–0.95) | <0.001 | 1.06 (1.03–1.09) | <0.001 | 0.85 (0.82–0.87) | <0.001 | 1.11 (1.08–1.14) | <0.001 | 1.04 (1.03–1.06) | <0.001 | 0.98 (0.97–0.99) | 0.04 |
| Also adjusted for obstetric practice factors[c] | 0.89 (0.86–0.91) | <0.001 | 1.01 (0.98–1.03) | 0.74 | 0.85 (0.82–0.87) | <0.001 | 1.09 (1.06–1.12) | <0.001 | 1.04 (1.03–1.06) | <0.001 | 0.98 (0.97–0.99) | 0.04 |
| Also adjusted for fetal/infant characteristics[d] | 0.91 (0.88–0.93) | <0.001 | 1.01 (0.99–1.04) | 0.80 | 0.86 (0.84–0.88) | <0.001 | 1.09 (1.07–1.13) | <0.001 | 1.04 (1.03–1.06) | <0.001 | 0.97 (0.96–0.98) | 0.03 |

\*Sequential adjustment was carried out by fitting a series of models with additional groups of factors added to each model in the sequence outlined above to quantify the contribution of each group of factors to CD trends over time.

[†]P-values represent significance of Wald chi-square test; the a priori level of statistical significance was set at a 2-sided p-value < 0.05.

[a]Maternal characteristics included maternal age, early-pregnancy BMI, smoking during pregnancy, and parity (only for Group 5 since Group 1 and 2a are restricted to nulliparous women).

[b]Maternal conditions included preeclampsia/eclampsia, preexisting diabetes, in vitro fertilization, and chronic hypertension.

[c]Obstetric practice factors included postterm delivery and epidural anesthesia (in Groups 1 and 2a only).

[d]Fetal/infant characteristics included position of the fetal head at delivery, infant birth weight, infant head circumference, and congenital anomaly.

BC, British Columbia; CD, cesarean delivery; CI, confidence interval; RR, rate ratio.

attenuated the difference in CD rates in BC versus Sweden in Groups 1, 2, 4, and 10 but not in Groups 5 and 8 (S22 Table).

Analyses restricted to individuals with no missing information on BMI and models with missing BMI values imputed yielded almost identical results for comparisons of CD rates between countries and within countries over time (S23 and S24 Tables).

## Discussion

We applied the Robson classification to 1,951,984 deliveries between 2004 and 2016 in Sweden and BC, Canada. The effect of controlling for maternal, fetal, and obstetric factors on the spatial comparisons differed by Robson group and ranged from no effect (e.g., among deliveries with breech presentation; Groups 6 and 7) to explaining up to 5.2% of the absolute 8.6% difference in CD rates in BC versus Sweden in Group 2 (nulliparous women with a term, cephalic fetus with induced labor, or a CD without labor). Nonetheless, wide spatial differences in CD rates persisted even after adjustment, especially in nulliparous women with a term, cephalic fetus with spontaneous labor, and women with a previous CD (Robson Groups 1 and 5, respectively). Adjustment for maternal, fetal, and obstetric practice factors explained a substantial fraction of the temporal change in CD rates in BC in Robson Groups 1 and 2a but had little impact on the temporal changes in CD rates observed among specific Robson groups in Sweden (such as Groups 2a and 5).

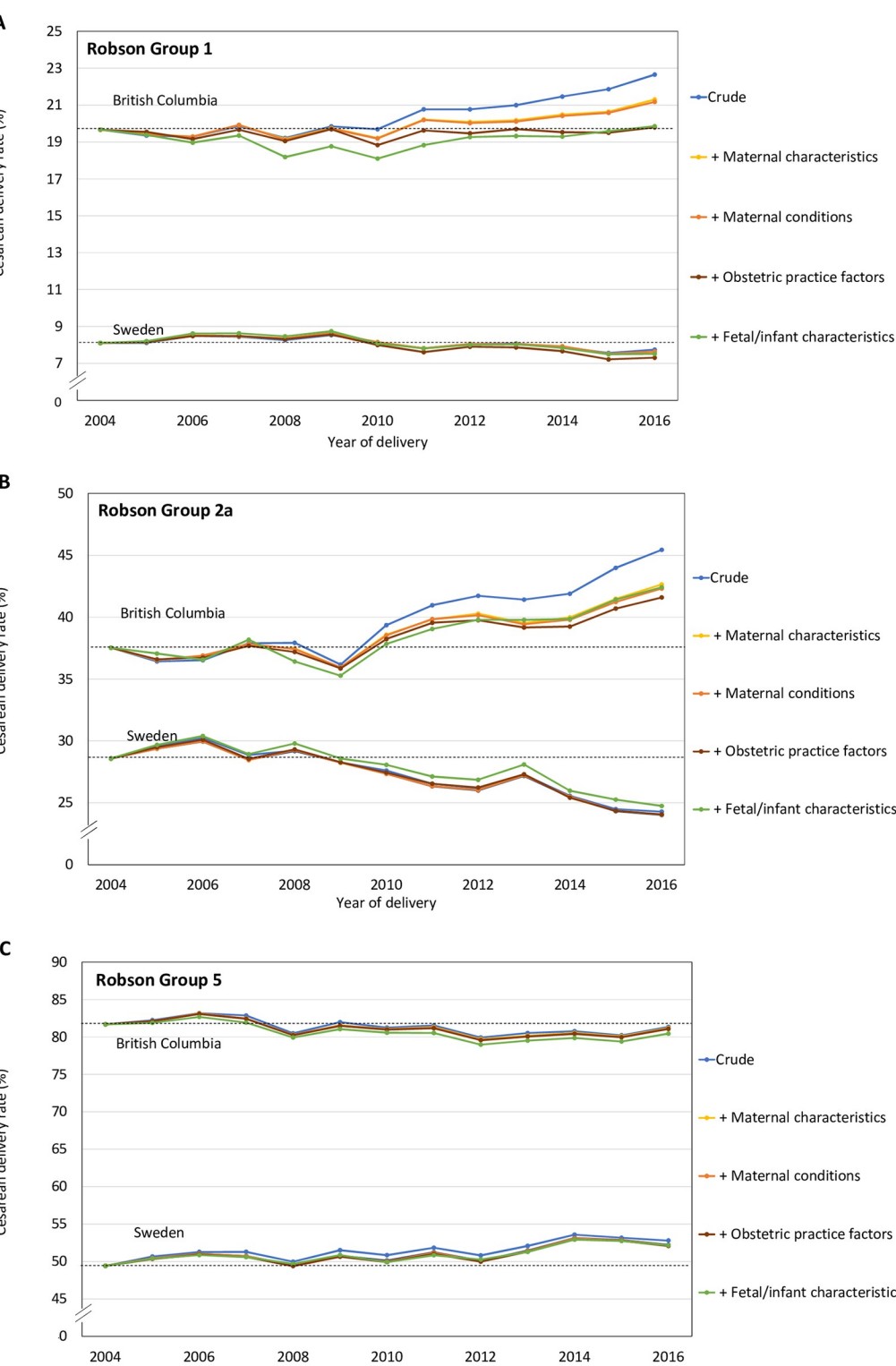

**Fig 4. Observed rates of CD and rates adjusted sequentially for changes in maternal characteristics, maternal conditions, obstetric practice, and fetal/infant characteristics among women in Robson Group 1 (panel A), Robson Group 2a (panel B), and Robson Group 5 (panel C) in Sweden and BC, Canada, 2004–2016.** Note that the y axis scales differ for the presentations of Robson Groups 1, 2a, and 5. BC, British Columbia; CD, cesarean delivery.

Several studies of obstetric populations in high-income settings have found that Robson Groups 1, 2a, and 5 contribute the most to overall CD rates [3–6], and the distribution of deliveries across Robson groups in both Sweden and BC is consistent with expected relative group sizes [33]. To our knowledge, there are no studies that have quantified international variation in CD rates using the Robson classification with and without adjustment for differences in maternal, fetal, and obstetric practice factors. One analysis that compared CD rates across 4 Nordic countries using the Robson classification adjusted solely for maternal age and did not report the variation in crude and adjusted estimates of CD [5]. The few studies that have combined Robson stratification and regression to estimate temporal variation in CD rates have reported mixed results. Similar to our findings, attenuation in CD rates over time after case-mix adjustment was observed among women in Robson Groups 1 and 3 in analyses from northern Italy [17,18]. On the other hand, a population-based study in France found that an increase in CD in Robson Group 1 from 2003 to 2010 did not change after adjustment for risk factors for CD, although this may be explained by limitations in available information as the models did not include BMI, maternal comorbidity, or fetal characteristics [19].

The strengths of our study include the ability to examine a large number of deliveries owing to the use of 13 years of data from population-based birth registers in 2 countries. In addition, we were able to include important determinants of CD in our analyses, such as maternal comorbidity, maternal BMI, fetal macrosomia, and position of the fetal head at delivery that have been excluded in previous analyses. The data sources used have been shown to be accurate in validation studies [24,25], and the proportion of women left uncategorized by Robson classification due to missing data was low (<1.0%).

The limitations of our study include potential data transcription errors and omissions in coding, which are inevitable in large health databases. Further, 13.8% of women (7.9% in Sweden and 28.3% in BC) in our study had missing values for early-pregnancy BMI and we included these women in our main models using a missing indicator approach. However, complete-case analyses (excluding cases with missing BMI values) and analyses including imputed values for missing BMI yielded similar results. Since we do not know if the BMI distribution was different among those with missing versus available BMI information, it is challenging to speculate on the direction and magnitude of bias introduced by missing data for BMI [34]. Confirmation of our findings in a cohort with complete BMI data would be valuable. The relatively large size of Group 4b (multiparous women with prelabor CD) in Sweden may indicate misclassification of women with a previous CD (i.e., they should have been assigned to Group 5); alternatively, this may indicate higher rates of CD on maternal request due to a variety of reasons, such as previously traumatic or prolonged labor. Finally, we were not able to account for important intrapartum details such as indication for CD, duration of labor, details regarding oxytocin augmentation, and fetal surveillance because our data sources lacked such information.

We quantified the crude and adjusted temporal trend in CD in a commonly used subdivision of Robson Group 2 restricted to women who had an induction (Group 2a) to increase the homogeneity of this group before assessing the contribution of maternal, fetal, and obstetric practice characteristics on temporal trends in CD. However, it is important to note that the analysis of any subdivision alone may be misleading without a consideration of the other subdivisions that compose the complete group (Group 2b in this case). In our analyses, the exclusion of Group 2b is unlikely to have had a substantial effect on temporal changes in the CD rate in Group 2 since the relative size of Group 2b was small (1.1% and 1.2% in Sweden and BC, respectively) and remained small throughout the study period. The relative size and high rate of CD in Group 2a are the reasons for this group's position as one of the 3 largest contributors to the overall CD rate in both Sweden and BC.

While our study explained spatiotemporal variations in CD rates, it did not address the perinatal and maternal morbidity and mortality that was caused or prevented by the observed differences in CD rates. Future studies are required to evaluate the differences in CD rates in each Robson group within the context of maternal, fetal, and infant morbidity and mortality. Our findings reinforce that, although often misinterpreted, the Robson classification was not meant to be an endpoint of CD comparison, but rather a starting point or initial framework within which determinants and outcomes of CD can be analyzed [10]. These results also highlight the importance of comprehensive and accurate perinatal data collection to ensure public health initiatives regarding CD rates are suitably evidence informed and prioritized. The maternal, obstetric practice, and fetal/infant characteristics included in our analyses were not exhaustive, and it is possible that the inclusion of other factors not represented in the Robson classification scheme (e.g., duration of labor, indication for CD) could have even larger impacts on CD rates.

The simplicity and efficiency of the Robson classification system has resulted in its endorsement by international health organizations and expansive uptake worldwide. However, our analyses show that maternal, fetal, and obstetric practice factors not included in the Robson classification explain a significant proportion of the spatiotemporal difference in CD rates in some Robson groups and should be incorporated into explanatory models evaluating CD rates in populations. Public health initiatives based on Robson-classified CD rates may lead to erroneous attributions of variation in CD rates to differences in obstetric practice without a comprehensive consideration of relevant maternal, obstetric, and fetal factors.

## Supporting information

**S1 Text. Study protocol.** Protocol for "Epidemiologic evaluation of cesarean delivery trends in Canada and Sweden" study.
(DOCX)

**S1 STROBE Checklist. Checklist of items that should be included in reports of observational studies.**
(DOCX)

**S1 Table. Assessment of quality of data, type of population, and cesarean delivery rates.** Steps defined by the World Health Organization to assess quality of data, type of population, and cesarean delivery rates using the Robson classification.
(DOCX)

**S2 Table. Crude and adjusted rates and odds ratios for cesarean delivery in British Columbia vs. Sweden by Robson group, 2004–2016.** To quantify comparisons of cesarean delivery rates in British Columbia vs. Sweden between Robson groups, we calculated odds ratios and 95% confidence intervals for Robson group-specific CD rates since the odds ratio (unlike the rate ratio) is not susceptible to artefactual constraints when the baseline CD rate is high.
(DOCX)

**S3 Table. Cesarean delivery rate by year of delivery and Robson group, Sweden, 2004–2016.** Robson group-specific cesarean delivery rates by year, percent change in cesarean delivery rates, and *p*-value for linear trend over the study period in Sweden.
(DOCX)

**S4 Table. Cesarean delivery rate by year of delivery and Robson group, British Columbia, Canada, 2004–2016.** Robson group-specific cesarean delivery rates by year, percent change

in cesarean delivery rates, and *p*-value for linear trend over the study period in British Columbia.
(DOCX)

**S5 Table. Maternal, obstetric practice, and fetal/infant characteristics in deliveries among women in Robson Group 1, Sweden and British Columbia, Canada, 2004–2016.** Distribution of determinants of cesarean delivery in Robson Group 1.
(DOCX)

**S6 Table. Maternal, obstetric practice, and fetal/infant characteristics in deliveries among women in Robson Group 2a, Sweden and British Columbia, Canada, 2004–2016.** Distribution of determinants of cesarean delivery in Robson Group 2a.
(DOCX)

**S7 Table. Maternal, obstetric practice, and fetal/infant characteristics in deliveries among women in Robson Group 2b, Sweden and British Columbia, Canada, 2004–2016.** Distribution of determinants of cesarean delivery in Robson Group 2b.
(DOCX)

**S8 Table. Maternal, obstetric practice, and fetal/infant characteristics in deliveries among women in Robson Group 3, Sweden and British Columbia, Canada, 2004–2016.** Distribution of determinants of cesarean delivery in Robson Group 3.
(DOCX)

**S9 Table. Maternal, obstetric practice, and fetal/infant characteristics in deliveries among women in Robson Group 4a, Sweden and British Columbia, Canada, 2004–2016.** Distribution of determinants of cesarean delivery in Robson Group 4a.
(DOCX)

**S10 Table. Maternal, obstetric practice, and fetal/infant characteristics in deliveries among women in Robson Group 4b, Sweden and British Columbia, Canada, 2004–2016.** Distribution of determinants of cesarean delivery in Robson Group 4b.
(DOCX)

**S11 Table. Maternal, obstetric practice, and fetal/infant characteristics in deliveries among women in Robson Group 5, Sweden and British Columbia, Canada, 2004–2016.** Distribution of determinants of cesarean delivery in Robson Group 5.
(DOCX)

**S12 Table. Maternal, obstetric practice, and fetal/infant characteristics in deliveries among women in Robson Group 6, Sweden and British Columbia, Canada, 2004–2016.** Distribution of determinants of cesarean delivery in Robson Group 6.
(DOCX)

**S13 Table. Maternal, obstetric practice, and fetal/infant characteristics in deliveries among women in Robson Group 7, Sweden and British Columbia, Canada, 2004–2016.** Distribution of determinants of cesarean delivery in Robson Group 7.
(DOCX)

**S14 Table. Maternal, obstetric practice, and fetal/infant characteristics in deliveries among women in Robson Group 8, Sweden and British Columbia, Canada, 2004–2016.** Distribution of determinants of cesarean delivery in Robson Group 8.
(DOCX)

**S15 Table. Maternal, obstetric practice, and fetal/infant characteristics in deliveries among women in Robson Group 9, Sweden and British Columbia, Canada, 2004–2016.** Distribution of determinants of cesarean delivery in Robson Group 9.
(DOCX)

**S16 Table. Maternal, obstetric practice, and fetal/infant characteristics in deliveries among women in Robson Group 10, Sweden and British Columbia, Canada, 2004–2016.** Distribution of determinants of cesarean delivery in Robson Group 10.
(DOCX)

**S17 Table. Frequency, proportion and rate ratio of maternal, obstetric practice, and fetal/infant characteristics among deliveries to women in Robson Group 1 in 2014–2016 vs. 2004–2007, Sweden and British Columbia.** Estimates of temporal trends in determinants of cesarean delivery in Robson Group 1.
(DOCX)

**S18 Table. Frequency, proportion and rate ratio of maternal, obstetric practice, and fetal/infant characteristics among deliveries to women in Robson Group 2a in 2014–2016 vs. 2004–2007, Sweden and British Columbia.** Estimates of temporal trends in determinants of cesarean delivery in Robson Group 2a.
(DOCX)

**S19 Table. Frequency, proportion and rate ratio of maternal, obstetric practice, and fetal/infant characteristics among deliveries to women in Robson Group 5 in 2014–2016 vs. 2004–2007, Sweden and British Columbia.** Estimates of temporal trends in determinants of cesarean delivery in Robson Group 5.
(DOCX)

**S20 Table. Crude and adjusted rate ratios for cesarean delivery in 2014–2016 vs. 2004–2007 among women in Robson Groups 1, 2a, and 5 after sequential adjustment for maternal characteristics, obstetric practice factors, and fetal/infant characteristics, Sweden and British Columbia, Canada.** Comparing the crude and adjusted differences in cesarean delivery rates by country and study period in Robson Groups 1, 2a, and 5.
(DOCX)

**S21 Table. Rate of cesarean delivery by Robson classification groups among women with midwifery care, British Columbia, Canada, 2004–2016.** Distribution of deliveries and cesarean deliveries restricted to women in British Columbia with midwifery-led maternity care.
(DOCX)

**S22 Table. Crude and adjusted rates and rate ratios for cesarean delivery among women with midwifery care, British Columbia, Canada, vs. Sweden, by Robson group, 2004–2016.** Comparing Robson group-specific cesarean delivery rates by country restricted to women in British Columbia with midwifery-led maternity care.
(DOCX)

**S23 Table. Crude and adjusted rates and rate ratios for cesarean delivery, British Columbia, Canada vs. Sweden, by Robson group, 2004–2016.** Comparing cesarean delivery rates in British Columbia vs. Sweden. (A) Excluding deliveries with missing values for body mass index and (B) using multiple imputation for missing body mass index values.
(DOCX)

**S24 Table. Crude and sequentially adjusted rate ratios of cesarean delivery rates in 2014–16 vs. 2004–2007 in Robson Groups 1, 2a, and 5, Sweden and British Columbia, Canada.** Comparing crude and adjusted temporal changes in the cesarean delivery rate by country. (A) Excluding deliveries with missing values for body mass index, and (B) using multiple imputation for missing body mass index values.
(DOCX)

**S1 Fig. Rate of preeclampsia/eclampsia by Robson group, Sweden and British Columbia, 2004–2016.** Proportion of women diagnosed with preeclampsia/eclampsia stratified by country and Robson group. Sweden is represented in the blue bars and BC is represented in the orange bars. The error bars indicate the 95% confidence interval. "RG" denotes Robson group, "Unk" denotes unknown.
(PDF)

**S2 Fig. Temporal trends in maternal characteristics, obstetric practice factors, and fetal/infant characteristics among women in Robson Group 1, Sweden and British Columbia, 2004–2016.** Changes in the frequency of determinants of cesarean delivery over the study period in Robson Group 1.
(PDF)

**S3 Fig. Temporal trends in maternal characteristics, obstetric practice factors, and fetal/infant characteristics among women in Robson Group 2a, Sweden and British Columbia, 2004–2016.** Changes in the frequency of determinants of cesarean delivery over the study period in Robson Group 2a.
(PDF)

**S4 Fig. Temporal trends in maternal characteristics, obstetric practice factors, and fetal/infant characteristics among women in Robson Group 5, Sweden and British Columbia, 2004–2016.** Changes in the frequency of determinants of cesarean delivery over the study period in Robson Group 5.
(PDF)

## Author Contributions

**Conceptualization:** Giulia M. Muraca, K.S. Joseph, Neda Razaz, Sarka Lisonkova, Olof Stephansson.

**Data curation:** Giulia M. Muraca, Neda Razaz, Linnea V. Ladfors, Sarka Lisonkova, Olof Stephansson.

**Formal analysis:** Giulia M. Muraca, K.S. Joseph.

**Funding acquisition:** Giulia M. Muraca, K.S. Joseph, Olof Stephansson.

**Investigation:** Giulia M. Muraca, Linnea V. Ladfors, Sarka Lisonkova.

**Methodology:** Giulia M. Muraca, K.S. Joseph, Neda Razaz, Linnea V. Ladfors, Sarka Lisonkova, Olof Stephansson.

**Project administration:** K.S. Joseph, Olof Stephansson.

**Resources:** Sarka Lisonkova, Olof Stephansson.

**Supervision:** K.S. Joseph, Olof Stephansson.

**Validation:** Linnea V. Ladfors.

**Visualization:** Giulia M. Muraca, Sarka Lisonkova, Olof Stephansson.

**Writing – original draft:** Giulia M. Muraca.

**Writing – review & editing:** Giulia M. Muraca, K.S. Joseph, Neda Razaz, Linnea V. Ladfors, Sarka Lisonkova, Olof Stephansson.

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
