## [Editor Report · Decision Letter 0]

24 Jan 2022

Dear Dr Muraca, 

Thank you for submitting your manuscript entitled "Adjusted comparisons of cesarean delivery rates between populations using the Robson classification: a population-based cohort study in Canada and Sweden" for consideration by PLOS Medicine.

Your manuscript has now been evaluated by the PLOS Medicine editorial staff and I am writing to let you know that we would like to send your submission out for external peer review.

Please re-submit your manuscript within two working days.

Kind regards,

Louise Gaynor-Brook, MBBS PhD

Senior Editor

PLOS Medicine

---

## [Decision Letter · Decision Letter 1]

23 Apr 2022

Dear Dr. Muraca,

Thank you very much for submitting your manuscript "Adjusted comparisons of cesarean delivery rates between populations using the Robson classification: a population-based cohort study in Canada and Sweden" (PMEDICINE-D-22-00231R1) for consideration at PLOS Medicine. 

Your paper was evaluated by five independent reviewers, including a statistical reviewer, and was discussed among all the editors here and with an academic editor with relevant expertise. The reviews are appended at the bottom of this email and any accompanying reviewer attachments can be seen via the link below:

[LINK]

In light of these reviews, we are not able to accept the manuscript for publication in the journal in its current form, but we would like to consider a revised version that addresses the reviewers' and editors' comments. Obviously we cannot make any decision about publication until we have seen the revised manuscript and your response, and we plan to seek re-review by one or more of the reviewers. 

We expect to receive your revised manuscript by May 09 2022 11:59PM. Please email us (plosmedicine@plos.org) if you have any questions or concerns.

We look forward to receiving your revised manuscript. 

Sincerely,

Louise Gaynor-Brook, MBBS PhD

PLOS Medicine

plosmedicine.org

General comments:

Throughout the paper, please adapt reference call-outs to the following style: "... gestational age (<37 vs ≥37 weeks) [8,9]." (noting the absence of spaces within the square brackets).

Data availability:

The Data Availability Statement (DAS) requires revision. If the data are owned by a third party but not freely available upon request, please state the owner of the data set and contact information for data requests (web or email address). Note that a study author cannot be the contact person for the data.

Title: Please revise your title according to PLOS Medicine's style. We suggest “Adjusted comparisons of cesarean delivery rates between populations using the Robson classification: A population-based cohort study in Canada and Sweden, 2004-2016” or similar

Abstract:

Please structure your abstract using the PLOS Medicine headings (Background, Methods and Findings, Conclusions).

Abstract Background: The final sentence should clearly state the study question.

Line 36 - ‘cesarean delivery’ is duplicated

Abstract Methods and Findings:

Please provide brief demographic details of the study population (e.g. fetal sex, maternal age, ethnicity, etc). Please also specify that data used in the analysis are from January 1, 2004 and December 31, 2016

Please define CD and BC at first use 

Please define Group 2

Please include the most important variables that are adjusted for in the analyses.

Please include the actual amounts and/or absolute risk(s) of relevant outcomes, not just RRs.

In the last sentence of the Abstract Methods and Findings section, please describe 2-3 of the main limitations of the study's methodology.

Abstract Conclusions:

Please begin your Abstract Conclusions with "In this study, we observed ..." or similar, to summarize the main findings from your study, without overstating your conclusions. Please emphasize what is new and address the implications of your study, being careful to avoid assertions of primacy. 

Author Summary:

In the final bullet point of ‘What Do These Findings Mean?’, please describe the main limitations of the study in non-technical language.

Methods:

Did your study have a prospective protocol or analysis plan? Please state this (either way) early in the Methods section. If a prospective analysis plan (from your funding proposal, IRB or other ethics committee submission, study protocol, or other planning document written before analyzing the data) was used in designing the study, please include the relevant prospectively written document with your revised manuscript as a Supporting Information file to be published alongside your study, and cite it in the Methods section. A legend for this file should be included at the end of your manuscript. If no such document exists, please make sure that the Methods section transparently describes when analyses were planned, and if/when reported analyses differed from those that were planned. Changes in the analysis-- including those made in response to peer review comments-- should be identified as such in the Methods section of the paper, with rationale. If a reported analysis was performed based on an interesting but unanticipated pattern in the data, please be clear that the analysis was data-driven.

Thank you for providing a STROBE checklist as Supporting Information. Please add the following statement, or similar, to the Methods: "This study is reported as per the Strengthening the Reporting of Observational Studies in Epidemiology (STROBE) guideline (S1 Checklist)." The STROBE guideline can be found here: http://www.equator-network.org/reporting-guidelines/strobe/ When completing the checklist, please use section and paragraph numbers, rather than page numbers which will likely no longer correspond to the appropriate sections after copy-editing.

Line 308 - Please replace "subject" with participant, patient, individual, or person.

Results: 

Please incorporate table S4 into the main paper as Table 1, to show the baseline characteristics of the study population.

Line 258 - Please revise use of ‘significant’ unless statistical analyses are provided to substantiate statistical significance 

Discussion:

Please rename your ‘Comment’ section to ‘Discussion’, and remove all subheadings e.g. ‘Principal findings’ and ‘Conclusion’

Please present and organize the Discussion as follows: a short, clear summary of the article's findings; what the study adds to existing research and where and why the results may differ from previous research; strengths and limitations of the study; implications and next steps for research, clinical practice, and/or public policy; one-paragraph conclusion.

Tables:

Please define all abbreviations used in the table legend of each table.

References:

Please ensure that journal name abbreviations match those found in the National Center for Biotechnology Information (NCBI) databases (http://www.ncbi.nlm.nih.gov/nlmcatalog/journals), and are appropriately formatted and capitalised. Please also name 6 authors prior to ‘et al’.

Where website addresses are cited, please specify the date of access. 

Comments from the reviewers:

Reviewer #1: Thanks for the opportunity to review your manuscript. My role is as a statistical reviewer so my comments focus on the study design, data, and analysis presented in the manuscript. I have put general comments first, and followed these with queries relevant to specific section of the manuscript (with a line reference).

This study uses data pooled from national data from Sweden and province level data from BC in Canada (from the birth registration systems). This manuscript evaluates if the standard for classification of deliveries used for comparing caesarean delivery rates can be improved by considering other maternal/fetal/health service characteristics that could explain variation in CDR. The study is large (~2 milliion deliveries) and aside from a few variables the data appears fairly complete. There are some queries below but the manuscript is written clearly and addresses an interesting question.

My background is not in working in perinatal epidemiology, and I was curious about how much of the overall rates in CD appear to come from a few specific Robson groups. Does the higher rates reflect a genuine healthcare need for these mothers, or is it from failure of the healthcare services to avoid an unnecessary CD? (this is question much more motivated by my own interest rather than a need to change the manuscript) 

One thing I think needs some rewording is the description of shift in RR after including additional covariates in the models. Typically ICC (and changes in) would be used a metric of variation and shift in unexplained variation, but to my knowledge this cannot be done with the modified Poisson regression models used. I think the approach of examining how the estimated RRs shift with additional adjustment is reasonable - but I think would be better simply described as absolute shift in estimated rate ratio to the specific Robson group/countries. 

L55. I am not familiar with the 'sequential' terminology - is this an extension of the Generalised Linear Model? 

L140. Is this extension of the Robson classification routinely used? 

L155. Are these comorbidity categories examples of the ones used, or the complete set?

L166. I have used GEE procedures with the modified Poisson regression to get estimates of rate ratios/relative risk (i.e. the robust standard error), in these cases each row/observation of data was independent so the independent working structure matched the data. If there are multiple deliveries per mother in the data, is an independent working structure realistic, i.e. compared to an exchangeable/CS structure? 

L171. This does assume that there is no correlation of missing BMI status with other covariates. This is assessed later on (seems to be fine), I would describe this approach as 'missing as indicator' in the methods. I would also specifically say you completed a complete-case analysis with respect to the other variables. I wasn't clear how many from each country were excluded on this basis - it would be helpful to have this detail in a flow diagram describing how many deliveries were initially included and then what remained after any exclusions for missing data. Was there variation in rates of missingness (in BMI and the other variables) over time?

L208. How did you decide on the selected number of imputations?

L286. Just checking the estimate and 95% CI here: 1.01 (0.99-1.04). This looks a bit odd - maybe just from rounding? 

Appendix - Table S23. Why did some of the adjusted models exclude particular variables? Because of small cell sizes?

Reviewer #2: Thanks for opportunity to review this paper.

* This is a novel and well-conducted analysis of retrospective birth data in 2 countries. The analysis addresses a well-known thorny issue in Robson-based analyses of CS usage. The methods are well-described.

* Did the authors explore the possibility of differences in the validity/reliability of the 2 datasets, particularly for Robson variables, which might have contributed to inter-country differences? 

* Line 158 - The consideration of 2 obstetric practice-related factors (Epidural anesthesia and post-term delivery) seemed somewhat brief to me and a potential major limitation. A number of practice or care-setting-related variables have been shown to be associated with CS rates, including but not limited to level of care (secondary vs tertiary), private vs public, antenatal care provision/provider, presence/absence of labour companion, continuity of care provider, VBAC policies, mandatory second opinion and others. I note that presence/absence of midwife-led care is dealt with through sensitivity analysis. I appreciate several of these are probably not be available in the individual level data, but it is necessary for the authors to explain the rationale for selection of only these 2 variables, and why others were not included or considered. 

* 203 - The issue of BMI driving CS rates is one of significant debate. Some clarification on the timing of early pregnancy BMI measurement would be useful. Also, 28% seems quite high for a missing rate and I did wonder if these are missing at random or not, and whether multiple imputation is appropriate in this context. 

Reviewer #3: This paper reports the findings of an innovative approach to contribute to the understanding of the CD epidemiology that is traditionally done using the well recognized and globally validated Robson classification. The authors add maternal, fetal and obstetrical practice factors to the Robson classifications in Sweden and BC.

The paper is well written, well established methodology taken into account models of obstetrical care, and reports on a huge sample size over a 12 years period of time, analyzing population based data sets. The findings are important, significant and show the weight of mainly maternal factors adding value to the Robson classification for clinical practice.

One question however remains: the constant lower and stable CD rates in Sweden as compared to the higher and increasing rates in BC can not be fully explained by differences in maternal or obstet practice factors. Did the authors have the opportunity to compare the CD rates in BC between midwifery and ob/gyn led models? or financial/ cost factors in the health care models of both sites?

Reviewer #4: This is a very interesting paper of almost 2 million deliveries between 2004 and 2016 in Sweden and BC Canada. It significantly adds to the literature. As the authors discuss, there has been a significant amount of work published on the Robson Ten Group Classification system, but the effect of controlling for maternal, obstetric and fetal factors the way the authors do in this work is novel.

I have one comment in relation to lines 218 - 220

"The rate of CD was higher in BC compared with Sweden in all Robson Groups except in Group 4 (parous women with a single, cephalic fetus at term gestation with an induction or CD prior to

labour) and among deliveries with a transverse/oblique-presenting fetus (Group 9)"

Group 9 is often used as a quality control mechanism for the data and the CS rate in this group should be 100%. Therefore, it may be reflective of the quality of the data collection rather than an actual difference itself. 

Notwithstanding this minor point, I believe that this work is very worthy of publication and gives a very clear insight into the significant spatio-temporal differences in certain Robson groups. 

Reviewer #5: Thank you for the opportunity to review your paper and i appreciate all the hard work that has gone into it.

I would like to make the following points. 

The TGCS has never been intended to be anything more than a starting point and a structure within which all other epidemiological variables, differences in practice and events and outcomes are analysed. I am not sure that the readers of the paper will understand that when you discuss it in the paper. 

I would always start by using the standard TGCS table first. There are a number of reasons why this is strongly advised

I get the impression from reading it that you have used data that is available rather than comment on the unavailability/quality of important information that should be there and if not that is the surely the big problem? So the TGCS structure is accurately collected (less then 1% of women could not be classified) but no real information on perinatal (in particular neonatal) outcomes or labour and delivery practice. Even BMI was not available in 28 % of women in BC.

There was no information on oxytocin rates, indications for CS or indications for pre-labour CS

I feel that there is a simple message from your TGCS data and that is that there are significant differences in CS rates between Sweden and BC. There are also more conclusions/hypotheses that you could have made from the raw TGCS table but you did not and i would be happy to help with that. With a good classification system there are only 3 reasons why there is a difference in either the sizes of the groups or the incidence of events within the groups. They are data quality, significant epidemiological variables and lastly differences in practice. By far the most common reason in perinatal data is data quality (including definitions)

While i absolutely agree with you that that the TGCS is only a structure and to make further conclusions in validating the difference between CS rates you need more information I am not sure with the limited data that you have presented that we are any further forward. 

The real point that your paper shows is how poor we all are in collecting routine perinatal data either because of a lack of resources or/and discipline together with a complete lack and agreement of further classifications. For example indications for CSs and inductions. Also your comment 369-371 supports this referring to lack of information on oxytocin, length of labour and fetal surveillance.

As far as the temporal and spatial differences in CS rates it is clear in the standard description of TGCS methodology that before considering the differences in CS rates you must examine any changes in the sizes of the groups first over the same period of time. You made no comment on that and even changed the design of the TGCS table by placing the column indicating the sizes of the group after the column showing the CS rates in each of the groups. This is confusing to readers who use the TGCS when each new paper invents their own way of designing the TGCS table

These comments are meant to be constructive and i think you should rethink what the main conclusion of what all your hard work has really shown especially the deficiencies. I would be happy to help in any way i can

[LINK]

---

## [Decision Letter · Decision Letter 2]

23 Jun 2022

Dear Dr. Muraca,

Thank you very much for submitting your revised manuscript "Crude and adjusted comparisons of cesarean delivery rates using the Robson classification: A population-based cohort study in Canada and Sweden, 2004-2016" (PMEDICINE-D-22-00231R2) for consideration at PLOS Medicine. 

Your paper was evaluated by a senior editor and discussed among all the editors here. It was also discussed with an academic editor with relevant expertise, and sent to three of the original reviewers, including a statistical reviewer. The reviews are appended at the bottom of this email and any accompanying reviewer attachments can be seen via the link below:

[LINK]

In light of the remaining reviewer comments, I am afraid that we will not be able to accept the manuscript for publication in the journal in its current form. However we would like to consider another revised version that addresses the reviewers' and editors' comments. Obviously we cannot make any decision about publication until we have seen the revised manuscript and your response, and we may seek re-review by one or more of the reviewers. 

We expect to receive your revised manuscript by Jun 30 2022 11:59PM. Please email us (plosmedicine@plos.org) if you have any questions or concerns.

We look forward to receiving your revised manuscript. 

Sincerely,

Caitlin Moyer

Associate Editor

PLOS Medicine

on behalf of 

Louise Gaynor-Brook

Associate Editor

PLOS Medicine

1. Response to reviewers: Please completely address the remaining comments of Reviewer 1 and Reviewer 5. Please carefully consider and respond to the comments provided by Reviewer 5 below.

2. Abstract: Methods and Findings: Line 45: Please revise the sentence for grammar.

3. Abstract: Methods and Findings: Line 45-46: Please separately mention numbers of deliveries for Sweden and BC.

4. Abstract: Line 54: Please use “women with overweight/obesity” in this sentence.

5. Line 55-56: Please also provide the p for trend for changes over time in each country.

6. Abstract: Line 75-76: We suggest revising to “These findings suggest that incorporating these factors into explanatory models may be useful for evaluating cesarean delivery rates in populations.” or similar.

7. Abstract: Conclusions: Please provide 1 sentence of additional interpretation of the findings (e.g. what is the significance of the fact that adjustment for these factors explains different proportions of variation in cesarean delivery between Robson groups? What is the significance that temporal changes in rates can be explained by adjustment for some groups/for Canada?).

8. Author summary: Line 98-99: Please clarify this, as the results do not indicate changes in CD rates over time in Sweden.

9. Author summary: Line 101-102: Please be sure that the author summary points do not reproduce what is written in the Abstract.

10. Methods: Please specify the significance level used (e.g., P<0.05, two-sided) and the statistical tests used to derive p values reported.

11. Results: Please provide both 95% CIs and p values for all applicable results reported in the text.

12. Discussion: Line 374: Please clarify if this is 5.2% of the 8.6% difference in CD rate for group 2 between Canada and BC.

13. Discussion: Line 380: “...but had little impact on temporal changes in CD rates in Sweden.” This is somewhat confusing as the results did not indicate that there were significant temporal changes in CD rates in Sweden. Please clarify.

14. Discussion: Line 381: Please clarify if “industrialized settings” might be more accurately described as high income settings, or similar.

15. References: Please include punctuation after journal title abbreviations. Please remove the registered trademark symbol from reference 30. Please use the "Vancouver" style for reference formatting, and see our website for other reference guidelines https://journals.plos.org/plosmedicine/s/submission-guidelines#loc-references

16. Tables and Supporting Information tables: Please quantify the presentation of results with both 95% CIs and p values where applicable. When reporting p values, please use 2 decimal places for p = 0.01 or larger, and 3 decimal places if smaller. Please report as p<0.001 where applicable. Please check that all abbreviations are defined in legends (e.g. CD).

17. Supporting Information Tables and Figures: Please provide a legend for each table/figure.

18. S1 Table: Please use the "Vancouver" style for reference formatting, and see our website for other reference guidelines https://journals.plos.org/plosmedicine/s/submission-guidelines#loc-references

19. S20 Table: Please clarify the title of the table, as it appears that all years (2008-2016) are being compared to 2004-2007. Please also include p values.

Comments from the reviewers:

Reviewer #1: Thanks for the revised manuscript and responses to my original queries. Your explanation to my question about variation in CD rates was extremely enlightening and I can appreciate why this is not an easy question to answer. 

The rest of the changes to the manuscript cover my initial questions (apart from a few minor points below) and I think you have given a fair assessment of the limitations with the missing BMI values. Apart the BMI the amount of missing data is a fairly small fraction and it would take a very unusual missingness mechanism to affect the main results of the study.

If using the 'sequential' term throughout the manuscript I would provide the explanation about what exactly you did (i.e. fitting a series of models with more covariates that could mediate the relationships of interest) to the methods so it's clear for readers. There are some statistical estimation methods that use this term and so best to make sure there's no confusion.

For Table S3 could you just add an brief explanatory note about where there was no p-value for some of the time-series (i.e. 2b, where it stayed the same across time)?

For Tables S5-S16, I would suggest the term 'unbalanced' instead of 'significant' as the convention is for 'significant' to be used with null hypothesis statistical testing. 

Reviewer #2: Happy with these responses. I am impressed at the thoroughness of the response to reviewer letter, many thanks. 

Reviewer #5: I do not disagree with your conclusion that you need other data in addition to the TGCS to explain differences in the caesarean section. Everybody would agree with that even before your study

I do however disagree that the criteria you have selected prove that or indeed we should encourage clinicians to collect those criteria before more relevant intrapartum data is collected. (i find it extraordinary that you cannot get information on indications, oxytocin rates or fetal outcome)

As a clinician i would have liked to like to see in your paper the TGCS as you have presented and within each of the groups a detailed analysis of relevant events and outcomes including epidemiological data. I also would expect to see in a paper about differences in cs rates a comparison of the indications and also other fetal and maternal outcomes. That way i would be able to compare my own data with yours.

There should also be some description of the clinical processes and how they differ in the two regions. 

I also advise caution using 2a as a separate group without 2b. in your dataset it may make no difference but in datasets where group 2b is relatively bigger you may draw inappropriate conclusions. It is not a Robson group and i would be happy to explain the reasons why that may be a problem

I really do believe that your paper shows the limitations of routine perinatal information at the present time and that should be the correct message from your paper nothing else

[LINK]

---

## [Editor Report · Decision Letter 3]

13 Jul 2022

Dear Dr. Muraca,

Thank you very much for re-submitting your manuscript "Crude and adjusted comparisons of cesarean delivery rates using the Robson classification: A population-based cohort study in Canada and Sweden, 2004-2016" (PMEDICINE-D-22-00231R3) for review by PLOS Medicine.

I have discussed the paper with my colleagues and the academic editor. I am pleased to say that provided the remaining editorial and production issues are dealt with we are planning to accept the paper for publication in the journal.

[LINK]

We look forward to receiving the revised manuscript by Jul 20 2022 11:59PM.   

Sincerely,

Caitlin Moyer, PhD

Associate Editor

PLOS Medicine

plosmedicine@plos.org

Requests from Editors:

1. Title Page: Please remove the funding and conflict of interest disclosure statements from the main text, and please be sure all information is submitted completely and accurately in the appropriate sections of the manuscript submission system.

2. Abstract: Methods and Findings: Line 59-63: Please also provide p values in addition to the 95% CIs reported for the main results comparing BC to Sweden.

3. Methods: Line 153: We suggest “data extraction” rather than “data manipulation” or similar.

4. Methods: Line 183-184: Please revise to “Temporal trends in the relative contribution of each Robson Group to the overall CD rate were also described.” if this was intended.

5. Results: Line 269: Please revise to “...while women of advanced maternal age (≥35 years) and women with overweight or obesity (≥25 kg/m2) constituted 23.5% and 32.4% of the study population…” or similar.

6. References: For reference 11, please change the journal to PLoS One. Please check that all references use the "Vancouver" style for reference formatting, and see our website for other reference guidelines: https://journals.plos.org/plosmedicine/s/submission-guidelines#locreferences

7. Page 29: Please remove the author contribution, declaration of interests, and data sharing sections from the main text. Please be sure all information is entered completely and accurately into the relevant sections of the manuscript submission system.

8. S20 Table and S24 Table: Please clarify “sequential adjustment” in the table legends along the lines of Reviewer 1’s point: ““The sequential approach was carried out by fitting a series of models with additional groups of factors added to each model in the sequence outlined above to quantify the contribution of each group of factors to CD trends over time.”

9. Figure 4: Please revise the y axis scales consistent across all panels, beginning with zero. If not feasible to do this, please show a break in the axis / include a note in the legend to call attention to the fact that the y axis scales differ for the presentations of Robson Groups 1, 2a, and 5.

10. STROBE Checklist: For item 22, please indicate the section as Funding or similar.

11. S1 Figure: Please include a legend. Please be sure to define what the error bars represent, and the abbreviation RG.

12. S2 Table: Please clarify if these are crude or adjusted OR. If applicable, please present both adjusted and unadjusted results. In the legend, please mention the factors adjusted for, if adjusted analyses were done.

[LINK]

---

## [Editor Report · Decision Letter 4]

15 Jul 2022

Dear Dr Muraca, 

On behalf of my colleagues and the Academic Editor, Gordon C. Smith, I am pleased to inform you that we have agreed to publish your manuscript "Crude and adjusted comparisons of cesarean delivery rates using the Robson classification: A population-based cohort study in Canada and Sweden, 2004-2016" (PMEDICINE-D-22-00231R4) in PLOS Medicine.

Please also address the following editorial request:

-Page 1: Paper presentation information: Please remove, or include as part of the acknowledgements.

PRESS

Sincerely, 

Caitlin Moyer, Ph.D. 

Associate Editor 

PLOS Medicine